# Embryonic Development and Growth Performance of the Tomato Hind Grouper (*Cephalopholis sonnerati*): A New Cultivated Aquaculture Species

**DOI:** 10.3390/ani15243655

**Published:** 2025-12-18

**Authors:** Yimeng Wang, Tangtang Ding, Yongsheng Tian, Dongqing Bai, Xinlu Jiao, Shihao Wang, Chunbai Zhang, Fengfan Yang, Linna Wang, Zhentong Li, Linlin Li, Yidan Xu, Yang Liu

**Affiliations:** 1Fisheries College, Tianjin Agricultural University, Tianjin 300392, China; 18222630398@163.com (Y.W.); dingtangtang0928@163.com (T.D.); baidongqing@tjau.edu.cn (D.B.); 15612990136@163.com (X.J.); 2State Key Laboratory of Mariculture Biobreeding and Sustainable Goods, Yellow Sea Fisheries Research Institute, Chinese Academy of Fishery Sciences, Qingdao 266071, China; tianys@ysfri.ac.cn (Y.T.); wangsh9899@163.com (S.W.); zhangchunbai204@163.com (C.Z.); yangfengfan2024@163.com (F.Y.); wangln@ysfri.ac.cn (L.W.); lizt@ysfri.ac.cn (Z.L.); lill@ysfri.ac.cn (L.L.); xuyidan7@126.com (Y.X.); 3Key Laboratory for Sustainable Development of Marine Fisheries, Yellow Sea Fisheries Research Institute, Ministry of Agriculture and Rural Affairs, Qingdao 266071, China; 4Hainan Innovation Research Institute, Chinese Academy of Fishery Sciences, Sanya 572025, China

**Keywords:** *Cephalopholis sonnerati*, embryonic development, feeding protocol, growth performance, variety breeding

## Abstract

The article conducted a systematic study on the growth and development characteristics of a newly cultivated *Cephalopholis sonnerati* species throughout the entire cultivation period.

## 1. Introduction

Groupers (*Epinephelinae*) are commercially important marine fish in China, with an annual aquaculture production of 241,500 tons and a market value of 30 billion CNY [1]. Grouper farming mainly focuses on hybrids, comprising more than 80% of the market due to their faster growth rates, disease resistance, and stress tolerance [2]. However, the development and use of high value purebred grouper species in aquaculture remain limited. As groupers are highly diverse, with over 160 species, current genetic improvement efforts still largely depend on crossbreeding, with the three newly approved grouper varieties all being hybrids. To date, there is a lack of research on aquaculture development and breeding of valuable purebred grouper species [3].

The tomato hind grouper *Cephalopholis sonnerati*, part of the Serranidae family, is predominantly found in tropical and subtropical coral reef environments [4]. The body is red with dark brown patterns on the head and back, giving it a decorative appearance. *C. sonnerati* is a newly developed and valuable species in aquaculture, with a market price of 300 CNY per kilogram, which is five times higher than that of hybrid groupers, highlighting its considerable economic potential for farming. However, *C. sonnerati* is classified as a slow-growing grouper, reaching a maximum weight of 548.67 g at 15 months old. Therefore, studying its breeding biology and genetics, as well as improving its growth rate, is especially important. To date, two telomere-to-telomere haploid genomes of *C. sonnerati* have been assembled, measuring 1039.53 and 1039.91 Mb in length, with 23,270 and 23,184 identified protein-coding genes, respectively [5]. Genomic evolution studies have shown that *C. sonnerati* diverged from the common ancestor of the giant grouper (*Epinephelus lanceolatus*) and the Hong Kong grouper (*E. akaara*) approximately 41.7 million years ago, and that the genes that have significantly expanded are primarily involved in sensory system pathways [6]. Current research on *C. sonnerati* mainly focuses on viral and bacterial infection [7,8], spawning and reproductive behaviors [9], body color [10], and fisheries sustainability [11], while there is a notable lack of research related to its cultivation in aquaculture.

For any aquaculture species, studying embryonic development and larval morphology is fundamental to achieving successful artificial breeding and large-scale seed production. Embryonic development, as the crucial initial life stage, follows specific timing patterns, adapts to environmental conditions, and exhibits developmental synchrony, all of which directly impact fertilization rates, hatching success, and the survival rate of newly hatched larvae [4]. Additionally, the morphological features of larvae and juveniles are closely associated with important ecological and physiological changes in the species, such as transitioning from internal to external nutrition, adjusting to artificial feed, and evading predators [5]. Recently, embryonic and larval development studies have been conducted on various interspecific and intergeneric hybrid groupers, such as *E. bruneus*♀ × *E. lanceolatus*♂ [12], *Cromileptes altivelis*♀ × *E. lanceolatus*♂ [13], *E. fasciatus*♀ × *Plectropomus*♂ [14], and *E. fuscoguttatus*♀ × *P. leopardus*♂ [15]. These studies have evaluated and differentiated the performance of hybrids and their parent species by examining factors such as hatchability, growth performance, morphological traits, chromosome karyotypes, muscle nutrient composition, mitochondrial genomes, and transcriptional levels. It was also discovered that the Wnt signaling pathway plays a crucial role in dorsal–ventral axis formation during the embryonic development of hybrid groupers [15]. However, research on the embryonic development and regulatory mechanisms of the rare purebred *C. sonnerati* remains relatively underdeveloped. Furthermore, growth characteristics are vital for cultured fish species because they directly affect the farming cycle and overall production yield [16]. Therefore, this research centers on the newly cultivated *C. sonnerati*, systematically assessing its embryonic development, early larval morphology, and growth performance. This study will offer theoretical and technical support for artificial breeding of *C. sonnerati* and to advance germplasm improvement for the species in aquaculture.

## 2. Materials and Methods

### 2.1. Experimental Materials

#### 2.1.1. Fertilized Egg Acquisition and Incubation

The broodstock breeding, fertilization and incubation, and fry rearing of *C. sonnerati* were collaboratively conducted by Laizhou Mingbo Aquatic Product Co., Ltd. (Yantai, China), and Hainan Lanliang Technology Co., Ltd. (Sanya, China). Eight mature female *C. sonnerati* and four male *C. sonnerati* (all exhibiting sperm motility above 85%), all aged between four and five years, were chosen for mixed fertilization. Fertilization and embryonic development were systematically observed and documented. The female broodstock had an average weight of about 2.32 ± 0.37 kg and an average total length of 47.50 ± 1.99 cm. The male broodstock averaged 2.07 ± 0.25 kg in weight and approximately 45.77 ± 1.54 cm in total length. The sexually mature broodstock was selected and administered injections comprising LHRH-A3 at dosage of 15 μg/kg and HCG at dosage of 300 IU/kg (Hubei Tusuo Technology Co., Ltd., Wuhan, China) to induce maturation and spawning. Following 48 h of hormone treatment, eggs were manually extracted from the abdomens of females for artificial insemination. About 1,200,000 fertilized eggs (weighing 600 g) were evenly divided among three incubation tanks (15.0 m^3^). After 20 min, 300 floating fertilized eggs were gathered from each tank to determine the fertilization rate. This procedure was conducted three times. The fertilized eggs were kept in micro-aerated water with a temperature of about 24 °C and a salinity ranging from 28 to 30 ppt. Once the embryos reached the tail bud stage, the eggs were transferred to a nursery pond for further cultivation, with an egg density of 7.5 g/m^3^ and a water temperature of approximately 25 °C.

#### 2.1.2. Larval Rearing and Grow-Out

The larvae of *C. sonnerati* were artificially reared and grown under controlled conditions with a water temperature of 26.0–28.0 °C and dissolved oxygen levels above 6.0 mg/L. Before the yolk sac was absorbed (0 to 5 days), the water exchange rate was kept between 0.2 and 0.3 m^3^/h. From day 9, when the dorsal and ventral fin spines had developed, the water flow was gradually increased to 5 m^3^/h.

The feeding regimen was as follows: from day 0 to 4, before the larvae began feeding, concentrated chlorella (8 × 10^5^ cells/mL) was added to the water; from day 5 to 15, small rotifers (SS) were provided at a density of 6 to 8 individuals per mL; from day 16 to 23, they were fed L-type rotifers twice daily at 5 to 6 individuals per mL; from day 24 to 30, they were fed *Artemia* nauplii twice daily at 3 to 4 individuals per mL; after 30 days, an artificial compound feed was introduced (Figure 1).

After this feeding protocol the fry entered a phase of rapid growth, during which cannibalism became severe, cannibalism rate reaches 14%. Therefore, feeding increased during the day, which continued until more than two-thirds of the fry stopped eating during each feeding session, occurring 2 to 3 times per day. After 30 min of feeding, the workshop lights were turned off, and PVC pipes along with other shelters were placed in the ponds, reduce the rate of cannibalism to within 5%. The fry obtained through mixed fertilization were consistently sieved and sorted for artificial rearing.

To evaluate the growth performance and address the biological gap in artificial cultivation of *C. sonnerati*, this batch of fry was farmed and graded in the factory. After 15 months of cultivation, a total of 17,868 fish from different parent pairs were graded and cultured in 21 ponds, with a stocking density of 15 kg/m^3^. To compare growth performance, the fish in each pond were ranked by average weight, and the five ponds with the highest average weights and the five with the lowest were selected as the fast-growing and slow-growing groups, respectively.

### 2.2. Sampling and Observation

During embryonic development observation, starting from fertilization, 30 floating eggs were periodically removed from the incubation tanks and examined with an optical microscope (Olympus CX43, Yijingtong Optics Technology Co., Ltd., Shanghai, China) and the characteristics and timing of each developmental stage were documented. The timing for each stage was determined when two-thirds of the fertilized eggs had reached that particular phase.

To assess morphological development, 15 fry were randomly selected from each developmental stage for total length measurement and to observe morphologic changes using a dissecting microscope (ZTR6745, Chongqing Zhiwei Micro Optical Instrument Co., Ltd., Chongqing, China). This was carried out every 2 to 3 days from 0 to 35 days post-hatching (dph), every 5 days from 35 to 45 dph, and every 10 days from 45 to 65 dph.

After 15 months of rearing, fish from both the fast-growing and slow-growing groups were photographed and measured and the total length and body weight were compared.

### 2.3. Calculations Using Formulas

The growth performance of *C. sonnerati* was assessed using the coefficient of variation and condition factor indicators. The formulas used to calculate these indicators are as follows:CV = (SD/Mean) × 100%K = (Weight/Length^3^) × 100%

### 2.4. Statistics

All statistical analyses of the growth data were conducted using SPSS 27.0. Initially, the Kolmogorov–Smirnov test and Levene’s test were applied to assess Normality and Homoscedasticity. The data points were approximately aligned along the 45-degree line, suggesting that the data were normally distributed (*p* > 0.05). For the Homoscedasticity assessment, since the *p*-value exceeded 0.05, a one-way ANOVA followed by a post-hoc LSD test was chosen to evaluate the significance analysis between groups, with a significance level set at *p* < 0.05. The results were expressed as mean ± standard deviation and visualized using Origin Pro 2022.

## 3. Results

### 3.1. Embryonic Development Observation

Fertilized *C. sonnerati* eggs completed embryonic development within 22:55 h after fertilization (hAF) at a water temperature of 24.8 ± 0.7 °C. The diameters of the fertilized eggs and the single oil globule measured 0.87 ± 0.02 mm and 0.22 ± 0.01 mm, respectively. Fertilization and hatching rates were 88.67 ± 3.93% and 79.67 ± 7.55%, respectively. Embryonic development was categorized into seven stages: fertilized egg, cleavage, blastula, gastrula, neurula, organogenesis, and hatching; the timing and features of early embryonic development are detailed in Table 1.

#### 3.1.1. Cleavage Stage

The cleavage pattern of *C. sonnerati*, like that of other groupers, is discoidal (Figure 2a). At 00:24 hAF, the blastodisc formed, where it appeared as a cap-like protrusion (Figure 2b). At 00:44 hAF, the embryo entered the 2-cell stage where the fertilized egg divided into two equal cells (Figure 2c). At 01:08 hAF, the egg underwent its second division, reaching the 4-cell stage (Figure 2d) and subsequently reached the 8-cell stage (Figure 2e), the 16-cell stage (Figure 2f), the 32-cell stage (Figure 2g), and, finally, the 64-cell stage (Figure 2h) at 02:13 hAF. At 02:43 hAF, the cells became smaller and with each division, their numbers increased, arranging irregularly and entering the multicellular stage (Figure 2i). At 03:16 hAF, the cells became less distinct, with the entire cell mass taking on a spherical shape resembling a mulberry, marking the morula stage (Figure 2j). Overall, the cleavage period included 10 developmental stages from the fertilized egg to the morula stage.

#### 3.1.2. Blastula Stage

As cell division progressed, both the number and layers of cells steadily increased. At 03:53 hAF, a blastocoel developed between the embryo and the yolk, and the central region of the blastula bulged outward like a tall cap, indicating the high blastula stage (Figure 2k). Following this, the blastula gradually flattened, and by 05:13 hAF, the protruding area reached its lowest point, the blastodisc became flat, cell division became more refined, and cell density increased, marking the transition to the low blastula stage (Figure 2l).

#### 3.1.3. Gastrula Stage

Cell division continued as the blastocyst cells progressively extended toward the vegetal pole, enveloping it downward. At 06:33 hAF, ⅓ of the yolk was covered by the embryonic layer, marking the early gastrula stage, where a crescent-shaped embryonic shield was visible from the side view (Figure 2m). By 07:43 hAF, the embryonic layer had enveloped half of the yolk, indicating the mid-gastrula stage (Figure 2n). At 08:43 hAF, the embryonic layer covered three-quarters of the yolk, reaching the late gastrula stage where the embryonic shield gradually elongated, and the embryo began to take shape (Figure 2o).

#### 3.1.4. Neurula Stage

Cell division persisted as the embryo progressed from the gastrula to the neurula stage. At 09:03 hAF, the dorsal region of the embryo gradually thickened, resulting in the formation of the neural plate with a central cylindrical notochord. Here, the embryo took shape with a clearly defined outline, entering the embryonic formation phase (Figure 2p). Subsequently, the embryonic ring continued to fold inward until a small opening, called the blastopore, appeared. By 09:40 hAF the blastopore closure stage was reached whereby the blastula had fully closed (Figure 2q).

#### 3.1.5. Organogenesis Stage

At 11:29 hAF, a pair of protrusions forming optic vesicles appeared on both sides of the embryo’s anterior head, marking the onset of optic vesicle formation (Figure 2r). By 12:21 hAF, somite primordia emerged on both sides of the notochord, indicating the start of the somite formation stage (Figure 2s). At 13:27 hAF, a pair of optic vesicles developed behind those in the embryonic head, and the number of somites increased, signaling the optic vesicle formation stage (Figure 2t). At 14:44 hAF, an oval brain vesicle formed between the optic vesicles and somite numbers continued to increase, marking the brain vesicle formation stage (Figure 2u). By 16:11 hAF, the heart had formed whereby the embryo entered the heart formation stage (Figure 2v). At 19:17 hAF, a small portion of the embryonic tail began to separate from the yolk sac and fin folds appeared on both the dorsal and ventral sides, indicating the tail bud stage (Figure 2w). At 19:59 hAF, highly refractive transparent lenses were visible within the optic vesicles and similarly refractive transparent otoliths formed within the optic vesicles; the embryo also began to twitch, marking the lens formation stage (Figure 2x). Finally, at 20:41 hAF, the heart started to beat faintly and gradually stabilized, signifying the heartbeat stage of embryo development (Figure 2y).

#### 3.1.6. Hatching Stage

At 21:54 hAF, the embryo started twitching vigorously and frequently, entering the pre-hatching phase (Figure 2z) and by 22:31 hAF, the larvae began to break free headfirst from the oolemma while the tail continued to twitch vigorously, marking the hatching stage (Figure 2z1). At 22:55 hAF, more than half of the larvae had hatched, concluding the embryo developmental process (Figure 2z2).

### 3.2. Larvae Characteristics

At 1 dph, the larvae measured 2.09 ± 0.12 mm in total length. Their bodies were transparent with melanin evenly spread across the head and tail. A yolk sac was present in the abdomen, noticeably smaller than the day before. The mouth opening was not yet developed, the digestive tract was not clearly defined, and the anus was positioned in the middle to rear part of the body. The larvae hung upside down in the water column, were evenly dispersed throughout the rearing tank, and exhibited weak movement (Figure 3a).

At 4 dph, the body length had increased slightly, most of the yolk sac was absorbed,. the digestive tract had developed and thickened, the anus had opened, and the mouth opening had formed, marking a gradual shift from internal to external feeding. Melanin deposits were present near the eyes and the digestive tract, with eye spots clearly visible, and a melanin cluster was also present in the abdomen. The pectoral fins grew, its swimming ability had slightly improved with quicker movement in the tail, and light was avoided with larval gatherings occurring in shaded areas (Figure 3b).

At 7 dph, the digestive tract continued to develop and thicken, improving feeding ability where ingested food inside the digestive tract was clearly visible. The mouth opening had enlarged, and the snout strongly extended forward. The swimming ability was greatly improved, including evasive maneuvers, and the larvae mostly congregated in the middle water layer. Melanin deposits appeared on the outer skin around the spine, with the abdominal melanin accumulating upwards. The pectoral fins grew larger, and the spinal structure became more defined (Figure 3c).

At 9 dph, the larvae developed the onset of the second dorsal fin spine along with a pair of pelvic fin spines, which gradually extended from the body surface forming the initial trident shape. Melanin was deposited at the tips of these fin spines, with a small cluster also visible near the rear end of the anus on the underside. The mouth opening had expanded and was distinctly positioned on the upper side (Figure 3d).

At 11 dph, the larvae’s pigment spots at the rear end and abdomen became larger and the second dorsal fin spine and pelvic fin spine showed slight growth, with the second dorsal fin spine slightly shorter than the pelvic fin spine. The larvae now mainly congregated in the middle to lower water areas (Figure 3e).

At 13 dph, the amount of melanin in the larvae’s abdomen increased, making the internal organs more difficult to see. There was no notable change in the length of the second dorsal fin spine or the pair of pelvic fin spines (Figure 3f).

At 15 dph, the second dorsal and the pelvic fin spines showed significant growth with the second dorsal fin having grown slightly faster, now having a roughly equal length to the pelvic fin spines. The initial formation of the first dorsal fin spine appeared in front of the second spine, which was now barbed with serrated spines starting to develop on the spines (Figure 3g).

At 17 dph, the first dorsal fin spine had developed, and the initial formation of the third dorsal fin spine was visible. The second dorsal fin spine was now noticeably longer than the pelvic fin spine. The tip of the pelvic fin spine featured black, ribbon-like structures that may aid in swimming. Melanin clusters on the tail and the ends of the fin spines grew larger, with a small amount also appearing on the larvae’s head, back, and snout (see Figure 3h).

At 20 dph, the second dorsal fin and pelvic fin spines continued to grow, the third dorsal fin spine started to develop, and the onset formation of the other dorsal fins began to appear. The operculum became clearly defined and the tail vertebrae started to curve upwards. Melanin at the tips of the dorsal and pelvic fin spines began to spread toward their bases, melanin patches on the tail and the top of the head increased in size, and the head melanin was now arranged in spots (Figure 3i).

At 23 dph, the third dorsal fin spine lengthened, with 5 to 6 spines already formed on the dorsal fin. The first spine emerged on the anal fin, the tail fin continued to develop, and the fin rays were now clearly visible. The operculum became thicker and took on an arched shape, while the stomach appeared pear-shaped and darker in color (Figure 4a).

At 26 dph, the second dorsal fin spine had reached its full length while the pair of pelvic fin spines was still developing. There were 11 dorsal fin spines aligned in a row, the dorsal and anal fins were distinct from the caudal fin, and the upward curvature of the caudal vertebrae had increased (Figure 4b).

At 29 dph, the pelvic fin spines reached the maximum length, the melanin on the head and snout diminished, the head became less transparent, and development progresses (Figure 4c).

At 32 dph, the second dorsal and pelvic fin spines were noticeably shorter, with the spine tips gradually becoming pointed. Scales started to form on the body surface and were reflective on the juvenile fish. The head was fully developed, and the operculum had a curved shape with a light red coloration around it (Figure 4d).

At 35 dph, the juvenile fish had black spots at the base of the dorsal fin, with all fins now essentially fully formed. The muscles on both sides of the body became gradually less transparent, the melanin concentration in the tail started to lighten, and the melanin at the tip of the pelvic fin spine disappeared, transforming into a sharp barb. The region around the eye socket and the abdomen appeared light yellow, while the area near the operculum was light red (Figure 4e).

At 40 dph, the dorsal fin spines had pulled back into sharp barbs, with their length roughly matching that of the pelvic fin spines. The caudal peduncle appeared yellow, with the color gradually fading toward the head along the lateral line. The red area surrounding the operculum had expanded and darkened, muscles continued to become less transparent, and the fry were able to move more swiftly and swim actively in the water (Figure 4f).

At 45 dph, the second dorsal fin spines were still shortening but had not completely retracted, all fins were more developed, and the body coloration was more pronounced. The total body length increased significantly, indicating rapid growth. Scales gradually spread over the entire body, resembling those of fingerlings in shape (Figure 4g).

At 55 dph, the second dorsal and the pelvic fin spines had shortened to their minimum length. The yellow markings around the eye sockets had gradually faded and were replaced by red patches, with the red coloration on the head becoming more intense. The fingerlings tended to aggregate under shelter (Figure 4h).

At 65 dph, total fingerling length measured 3.5 ± 0.2 cm. The second dorsal and pelvic fin spines began to grow again, and the eyes became more prominent. The body displayed vivid colors with the head showing a dark red hue, the abdomen and caudal peduncle a deep yellow, and the back a silvery-gray shade. Interestingly, as the fish grows, the color of the patches on its head changes from yellow to red. The scales became opaquer, the fins remained transparent, and the capacity to produce mucus on the skin surface increased (Figure 4i).

### 3.3. Observations on Larval Feed Transition and Growth and Development

Early growth of *C. sonnerati* was closely connected to its feeding transition strategy (Figure 5). From hatching until 29 dph, the fry gradually shifted from internal nutrition to external feeding. During this time, their total body length increased steadily at an average rate of 0.20 mm per day, and individual variation began to emerge. After switching to artificial feed, the fry entered a rapid growth phase, with 35 dph marking a significant milestone. Here, their scales were fully developed, and their appearance resembled that of fingerlings. From here, the average daily growth rate increased to 0.85 cm, with simultaneous acceleration in body color differentiation, highlighting the growth advantage of individuals that accumulated red pigmentation on their heads. Between 55 and 65 dph, the fingerlings underwent a developmental process involving elongation, contraction, and regeneration of the second dorsal and pelvic fin spines. This was accompanied by changes in body coloration where fast-growing individuals obtained deepened red patches on their heads, while slower-growing ones retained yellow patches from the larval stage, resulting in a distinct phenotypic difference. These patterns of growth and development differentiation continued to intensify during later cultivation stages. Due to the continuous growth pattern, an exponential model was applied to fit the growth curve of *C. sonnerati*, with an R^2^ value of 0.954. The fitted equation is y = 0.1268e^0.1483x^, where y denotes the total length and x represents the age in days.

### 3.4. Evaluating Growth Performance

After 15 months of size-grading farming (Figure 6, Table 2), 30 fish were randomly chosen from each pond to measure their weight and length. When ranking the average weights of fish from each pond, it was observed that the largest *C. sonnerati* weighted 548.67 g, while the smallest was only 125.41 g, showing considerable growth difference within the population. The fast-growing group had an average weight of 457.12 ± 58.68 g, which was 2.90 times heavier than the slow-growing group of 157.86 ± 22.94 g. Additionally, the fast-growing fish had an average total length of 29.06 ± 1.23 cm and a body length of 23.66 ± 1.39 cm, which were 1.46 and 1.41 times longer than those of the slow-growing group, respectively. The condition factor was slightly higher in the fast-growing group compared with the slow-growing group, while the weight variation coefficient was lower in the fast-growing group (Table 2). After long-term cultivation, the overall survival rate was 90.53 ± 2.67%. Morphologically, *C. sonnerati* typically displayed a dark red body without distinct stripes, where the dorsal and ventral areas were clearly red, and the fins were generally darker. The adult fish had a pronounced arched back. The fast-growing fish had a robust, spindle-shaped body, while the slow-growing fish appeared slender. The comparative increased weight within the fast-growing group was mainly due to greater muscle and fat accumulation in the dorsal and ventral regions, likely driven by the significantly higher feeding intensity observed compared with the slow-growing group.

## 4. Discussion

### 4.1. Embryonic Development Evaluation

As an economically important coral reef species, understanding the early developmental traits of *C. sonnerati* is vital for conserving genetic resources and supporting artificial breeding of the species. Therefore, this study offers a comprehensive overview of the morphological transformations from embryo to fingerling, evaluates overall growth performance, highlights the ecological adaptability of its developmental strategy, and identifies key factors necessary for successful artificial cultivation of *C. sonnerati*.

The embryo of *C. sonnerati* displays typical characteristics of pelagic eggs, which help them disperse through surface ocean currents. The rate of embryonic development in groupers varies under different incubation temperatures. While the degree-hours (water temperature × number of hours passed) model accumulates the number of °C, this two-parameter model, by using the product of degree-hours, takes into account accelerations and decelerations in development at high and low temperatures and thus provides more accurate estimations of embryonic development [17]. In this study, at a water temperature of 24.8 ± 0.7 °C, the embryonic development of *C. sonnerati* was completed in 22 h and 55 min. The degree-hour of *C. sonnerati* is lower than that of *E. fuscoguttatus* [18], indicating faster development, but it is higher than that of *E. lanceolatus* [19] and *Plectropomus leopardus* [20]. *E. lanceolatus* and *P. leopardus* exhibit higher optimal development temperatures of 29 °C and 30.6 °C, respectively, whereas 24.8 °C is more suitable for *C. sonnerati* (Table 3) These observations suggest an inverse relationship between water temperature and embryonic development duration: higher temperatures result in shorter development times. Studies on the effect of temperature on *E. moara* embryonic development indicate that the optimal hatching temperature is between 22 and 24 °C; temperatures above this range reduce hatching rates and increase malformations [21]. Additionally, research on hybrid groupers showed that when hatching temperatures fall below 17 °C, the hatching rate of hybrid embryos from *E. moara* (female) × *E. septemfasciatus* (male) decreased as temperature drops, with malformation rates rising [22]. Conversely, temperatures above 25 °C also reduce hatching rates and increase malformations, with the ideal hatching temperature being 17 to 21 °C. This demonstrates that hatching time, survival rate, malformation rate, and water temperature during hatching are closely interconnected in fertilized grouper eggs. Generally, higher hatching temperatures shorten development time; however, extreme temperatures cause developmental stress, leading to increased mortality and deformities. Therefore, precise control of incubation water temperature is critical for shortening the incubation period, improving synchronization, and enhancing survival rates of artificially bred and reared groupers. Concurrently, maintaining appropriate water quality, salinity, and dissolved oxygen levels is also essential [23].

Like many other groupers, a distinctive feature in the early larval development of *C. sonnerati* is the formation and subsequent regression of the elongated second dorsal and paired pelvic fin spines, collectively known as the “tripartite spines” or “trident” [24]. In this study, spine protrusion started at 9 dph, reached its maximum length around 26–29 dph, significantly regressed from 32 dph onward, and began to regenerate by 65 dph, reaching a total length of 3.5 ± 0.2 cm, indicating the completion of metamorphosis development [25]. Mortality during this stage is a major factor limiting the large-scale farming of grouper fry [26]. In the hybrid grouper of *Epinephelus lanceolatus* (♀) × *E. tukula* (♂), the second dorsal spine regenerated at 47 days, with a total length of 2.46 ± 0.4 cm [27]. Similarly, the backcross grouper between female *E. moara* and male *E. moara* (♀) × *E. lanceolatus* (♂) showed regeneration by 60 dph, with a total length of 2.8 ± 0.2 cm [28]. Compared with these hybrids, the regeneration time in *C. sonnerati* is slightly delayed, which may enhance the survival rate by allowing for increased body size. This elaborate sequence of elongation, contraction, and potential regeneration represents specialized developmental unique to the early life stages of serranid fish such as groupers. The biological and ecological significance of this transient structure is multifaceted and is considered an evolutionary adaptation that enhances larval survival and dispersal in the pelagic environment. Combined with a relatively large oil globule, the spines modify larval hydrodynamics by increasing drag and potentially aiding flotation, helping the larvae remain in productive surface waters and facilitating oceanic dispersal, a crucial trait for reef-associated species with a pelagic larval phase. Furthermore, the elongated, often pigmented spines significantly increase the apparent size of the larvae, making them visually larger and more difficult to be ingested by small gape-limited predators. The spines may also make the larvae mechanically challenging to handle and swallow [29].

The dynamic development of these tripartite spines in *C. sonnerati* is a sophisticated evolutionary phenotype embodying a series of adaptive compromises to the conflicting demands of predator avoidance, oceanic dispersal, and the eventual benthic settlement. Understanding this specialized developmental mode provides deeper insights into the early life history strategies of groupers and underscores the importance of managing such critical developmental windows in artificial breeding programs [30].

### 4.2. Growth Performance Evaluation

Growth rate is a crucial economic characteristic in aquaculture fish farming as it directly influences the farming cycle and yield. It is important to note that in real farming conditions, it has been observed that as fingerlings develop, the color of their head patches changes from yellow to red, offering a quick and non-invasive method for early identification of fast-growing traits. For the full cycle of cultivation, the 15-month-old *C. sonnerati* reached a minimum and maximum body weight of 125.41 g and 548.67 g, respectively, categorizing it as a slow-growing grouper species. Among over 20 cultured grouper species, only *E. lanceolatus* and potato grouper (*E. tukula*) are classified as large, weighing over 100 kg, while the others are considered medium to small. Years of crossbreeding have demonstrated that growth rates of hybrid grouper offspring can surpass the parents [31]. For instance, one-year-old Yunlong grouper (*E. moara* female × *E. lanceolatus* male) weigh 700 g and grow more than twice as fast as the maternal *E. moara* [32]. Similarly, 15-month-old Jinhu grouper (*E. fuscoguttatus* female × *E. tukula* male) can reach 850 g, growing 103% faster than the maternal *E. fuscoguttatus* [33]. By crossbreeding, *C. sonnerati* can be hybridizedwith faster-growing species to develop new varieties exhibiting hybrid vigor, thereby enhancing growth rates and farming efficiency. To date, male *C. sonnerati* and female *E. fuscoguttatus* have been hybridized, resulting in offspring with superior growth traits. The average weight of three-month-old hybrids was 24.0 ± 3.6 g, 1.4 times that of the *E. fuscoguttatus*, offering theoretical support for future genetic improvement studies targeting growth traits in *C. sonnerati*.

This study also identified notable differences in body weight within the farmed *C. sonnerati* population, with the fast-growing group averaging 2.9 times the weight of the slow-growing group, demonstrating clear growth differentiation. This aligns with allometric growth characteristics seen in grouper aquaculture, although the extent of these differences is considerably greater than that observed in other hybrid grouper varieties. Research has shown that *P. leopardus* displays significant individual growth variations during artificial breeding, attributed to genetic differences [34]. Therefore, it is suggested that the growth disparities observed in *C. sonnerati* might link to genetic factors. However, current research has not established family structures, lacks pedigree records, and lacks verification of kinship tracing. Therefore, the contribution of genetic factors to the growth differences observed in the *C. sonnerati* requires further clarification through subsequent dedicated studies. Combining experimental observations with practical aquaculture experience, environmental factors exert a more direct and traceable driving force on growth differentiation. For example, severe cannibalism during the early fry stage allows slightly larger individuals to establish dominance hierarchies, capturing more feed resources and perpetuating a “survival of the fittest” differentiation cycle. Around 30 dph, during the crucial feed transition phase, individuals show considerable differences in how quickly they adapt to artificial feed, with a survival rate of 70% at this stage. Rapid adapters demonstrate higher feeding intensity and accelerated nutrient accumulation, consistent with the observation that “the rapid growth group exhibited significantly stronger feeding intensity than the slow group.” Additionally, micro-environmental heterogeneity in water temperature and dissolved oxygen levels within factory-based aquaculture [35], coupled with uneven distribution of shelter structures like PVC pipes [36], further exacerbates disparities in individual metabolic efficiency and stress states. Therefore, individuals that adapt more rapidly to artificial breeding gain an energy accumulation advantage, while slower-growing individuals may enter a negative cycle of poor feeding leading to stunted growth through low feed efficiency. Due to the large individual differences in growth rates, smaller fry are more vulnerable to predation by larger fry, reducing survival rates and potentially prolonging the aquaculture cycle resulting in inconsistent market timing and impacting economic returns. Studies have found that selected Chinese mitten crab (*Eriocheir sinensis*) show improved growth performance and immunity and disease-related physiological traits compared with unselected populations [37]. Additionally the growth rate of goldfish (*Carassius auratus*) also significantly increases after selection [38]. Conducting parental selection research is important to enhance the uniformity and survival rates of fry. Therefore, even after successful artificial breeding, *C. sonnerati* requires ongoing selection research to further improve the growth performance of this valuable grouper species.

## 5. Conclusions

This study systematically examined the embryonic development, morphological changes, and growth performance of *C. sonnerati* at various developmental stages. At a water temperature of 24.8 ± 0.7 °C, the embryos developed within 22 h 55 min, corresponding to 568.42 degree-hours. The newly hatched larvae measured 2.09 ± 0.12 mm in total length, and during the morphological changes in larvae and juveniles, the fish entered a rapid growth phase after transitioning to artificial feed at 30 dph. Graded rearing at 15 months revealed significant differences in population growth. Therefore, proper feed conversion and graded rearing can enhance growth performance. However, the importance of parental selection and genetic improvement research is emphasized to ensure uniformity and improved fingerling growth rates which will improve the cultivation of this species in the future. For this purpose, it is advised to carry out further research focused on selecting growth traits and identifying high-efficiency hybrid combinations for *C. sonnerati*.

## Figures and Tables

**Figure 1 animals-15-03655-f001:**
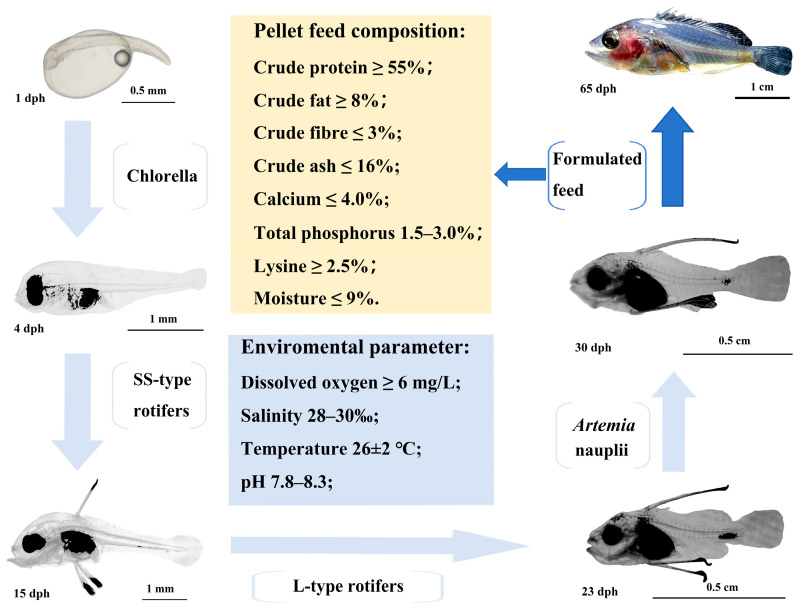
The feeding protocol of *C. sonnerati*.

**Figure 2 animals-15-03655-f002:**
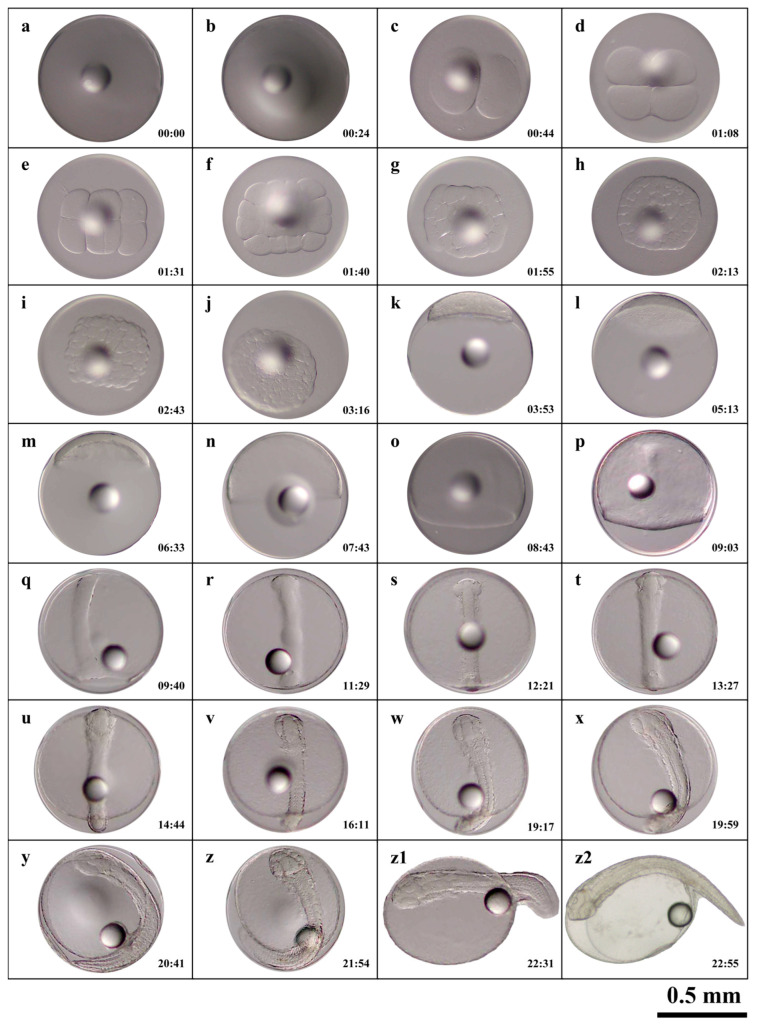
The embryonic development process of *C. sonnerati*: (**a**) fertilized egg, (**b**) blastodisc formation period, (**c**) 2-cell stage, (**d**) 4-cell stage, (**e**) 8-cell stage, (**f**) 16-cell stage, (**g**) 32-cell stage, (**h**) 64-cell stage, (**i**) multicellular stage, (**j**) morula stage, (**k**) high blastula stage, (**l**) low blastula stage, (**m**) early gastrula stage, (**n**) middle gastrula stage, (**o**) late gastrula stage, (**p**) embryo formation stage, (**q**) blastopore closure of stage, (**r**) optic vesicle formation stage, (**s**) somite formation stage, (**t**) otic vesicle formation stage, (**u**) brain vesicle formation stage, (**v**) heart formation stage, (**w**) tail-bud stage, (**x**) lens formation stage, (**y**) heartbeat stage, (**z**) pre-hatching stage, (**z1**) hatching stage, (**z2**) newly hatched larvae. The number in the lower right corner represents the hAF with a scale of 0.5 mm.

**Figure 3 animals-15-03655-f003:**
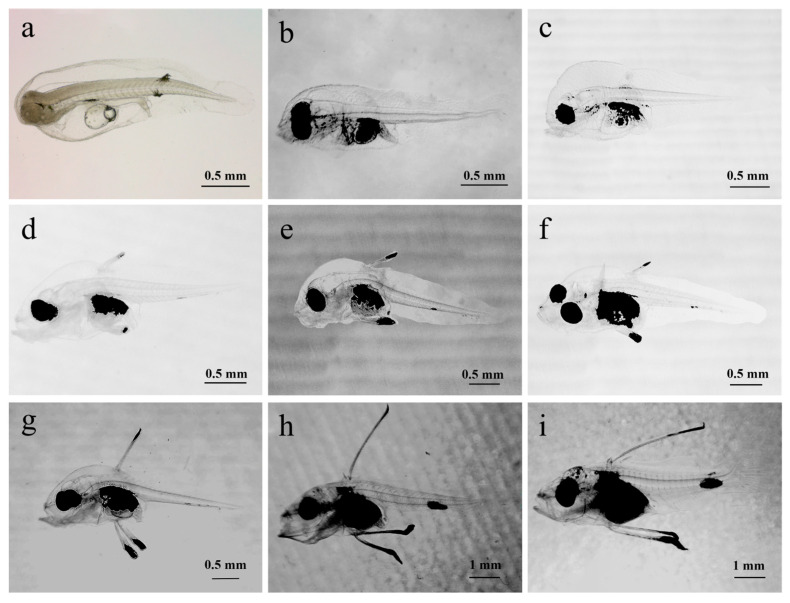
Morphological development of larvae, juvenile fish, and fingerlings of *C. sonnerati*: (**a**) larvae at 1 dph, (**b**) larvae at 4 dph, (**c**) larvae at 7 dph, (**d**) larvae at 9 dph, (**e**) larvae at 11 dph, (**f**) larvae at 13 dph, (**g**) larvae at 15 dph, (**h**) larvae at 17 dph, (**i**) larvae at 20 dph.

**Figure 4 animals-15-03655-f004:**
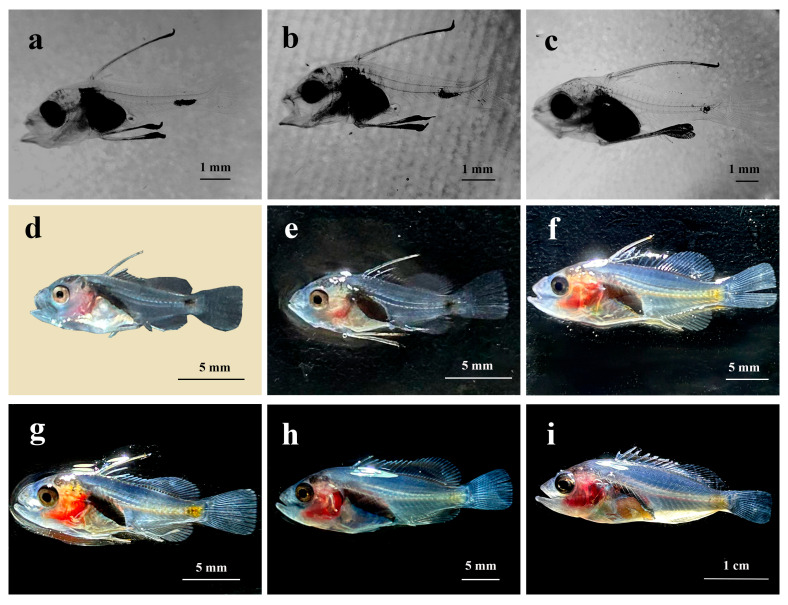
Morphological development of larvae, juvenile fish and fingerlings of *C. sonnerati*: (**a**) larvae at 23 dph, (**b**) larvae at 26 dph, (**c**) larvae at 29 dph, (**d**) juvenile fish at 32 dph, (**e**) juvenile fish at 35 dph, (**f**) juvenile fish at 40 dph, (**g**) juvenile fish at 45 dph, (**h**) fingerlings at 55 dph, (**i**) fingerlings at 65 dph.

**Figure 5 animals-15-03655-f005:**
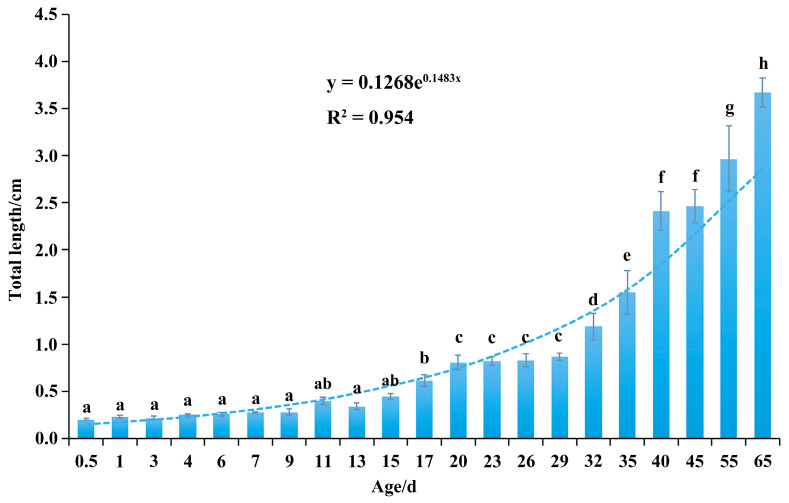
The relationship between the total length and age in *C. sonnerati* larvae. Different letters represent significant differences (*p* < 0.05).

**Figure 6 animals-15-03655-f006:**
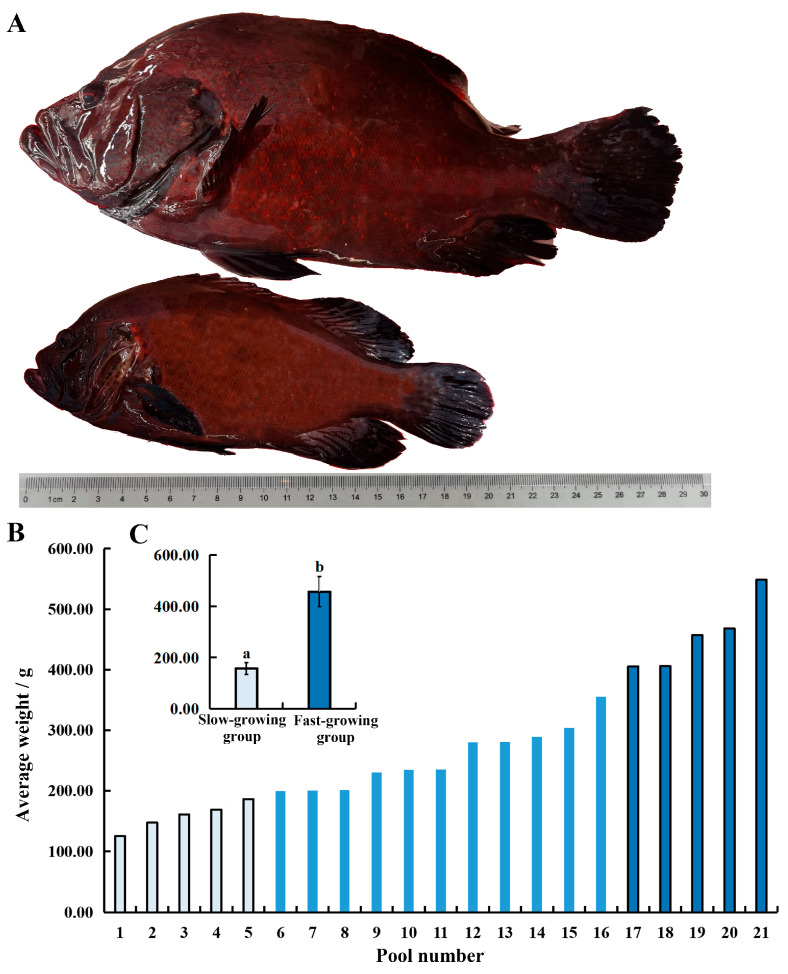
Analysis of the growth difference of 15-month-old *C. sonnerati*: (**A**) body length comparison, (**B**,**C**) body weight comparison. Different letters represent significant differences (*p* < 0.05).

**Table 1 animals-15-03655-t001:** The timeline and characteristics of embryonic development of *C. sonnerati*.

Embryonic Developmental Stage	Detailed Developmental Stage	Primary Characteristics	hAF
Fertilized egg	Fertilized egg	Spherical, with 1 oil globule	00:00
Cleavage stage	Blastodisc formation period	Blastodisc appears as a cap-like protuberance	00:24
	2-cell stage	1st cleavage, forming 2 equivalent cells	00:44
	4-cell stage	2nd cleavage, forming 4 equivalent cells	01:08
	8-cell stage	3rd cleavage, forming 8 equivalent cells	01:31
	16-cell stage	4th cleavage, forming 16 cells	01:40
	32-cell stage	5th cleavage, forming 32 cells	01:55
	64 cell stage	6th cleavage, forming 64 cells of unequal size and irregular arrangement	02:13
	Multicellular stage	Continued division, increasing cell count	02:43
	Morula stage	Cells accumulate in multiple layers, appearing round, resembling a mulberry	03:16
Blastula stage	High blastula stage	The blastoderm is high and concentrated, appearing like a high hat in lateral view	03:53
	Low blastula stage	The blastoderm becomes lower, cells are preparing to envelop the vegetal pole	05:13
Gastrula stage	Early gastrula stage	The blastoderm covers 1/3 of the yolk, and the embryonic shield is visible laterally	06:33
	Middle gastrula stage	The blastoderm covers 1/2 of the yolk	07:43
	Late gastrula stage	The germ layers enclose 3/4 of the yolk, embryonic shield becomes elongated, and the embryo is forming	08:43
Neurula stage	Embryo formation stage	Embryo formation, distinct outline	09:03
	Blastopore closure stage	Epiboly, blastopore completely closed	09:40
Organogenesis stage	Optic vesicle formation stage	A pair of optic vesicles appears in the embryonic head	11:29
	Somite formation stage	Somites appear in the middle of the embryo	12:21
	Otic vesicle formation stage	A pair of optic vesicles appear posterior to the optic vesicles in the head	13:27
	Brain vesicle formation stage	Brain vesicles appear between the two optic vesicles	14:44
	Heart formation stage	The heart forms ventrally, with a clear outline	16:11
	Tail bud stage	The caudal part of the embryo begins to separate from the yolk sac	19:17
	Lens formation stage	Lens appears in the embryo’s eyes	19:59
	Heartbeat stage	Heart begins to beat faintly, then gradually stabilizes	20:41
Hatching stage	Pre-hatching stage	Embryo twitches violently	21:54
	Hatching stage	Head first out of membrane	22:31
	Newly hatched larvae	Larvae hatch out of membrane	22:55

**Table 2 animals-15-03655-t002:** Comparison of growth difference between fast- and slow-growing groups.

Group	Body Weight (g)	Total Length (cm)	Body Length (cm)	Condition Factor	Coefficient of Variation for Body Weight
Fast-growing	457.12 ± 58.68 ^b^	29.06 ± 1.23 ^b^	23.66 ± 1.39 ^b^	3.45 ± 0.23	12.84
Slow-growing	157.86 ± 22.94 ^a^	19.92 ± 0.94 ^a^	16.82 ± 1.33 ^a^	3.32 ± 0.30	14.53

Note: Different superscript letters represent significant differences (*p* < 0.05).

**Table 3 animals-15-03655-t003:** Comparison of degree-hours among different grouper species.

Species	Incubation Water Temperature (°C)	Hatching Time (Hours)	Degree-Hours
*C. sonnerati*	24.8	22.92	568.42
*E. lanceolatus*	29.0	18.50	536.50
*E. fuscoguttatus*	27.3	22.00	600.60
*P. leopardus*	30.6	16.53	505.82

## Data Availability

The original contributions presented in this study are included in the article. Further inquiries can be directed to the corresponding author.

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
