# Peer review of "Embryonic Development and Growth Performance of the Tomato Hind Grouper (Cephalopholis sonnerati): A New Cultivated Aquaculture Species"

_animals, 2025, doi:10.3390/ani15243655_

Round 1
Reviewer 1 Report
Comments and Suggestions for Authors
- The Abstract misses the key finding such as the correlation between fast growth in red patches and slow growth in yellow patches on the head. Please incorporate this distinctive phenotypic marker into the Abstract as it is a major practical finding.
- The fertilization rate was derived from a small sample size of only 300 eggs. Please justify the representativeness of this subsample, if a larger population was used for the final rate.
- The authors found that sonnerati is slow-growing, the Introduction needs to strengthen the rationale for its culture.
- There are inconsistencies in naming, such as "Somite formation stage" vs. "Muscle burl stage," between Table 1 and Figure 2. Ensure a consistent set of standardized scientific terminology is used across all figures, tables, text, etc. for the better transparency.
- The discussion correctly notes the initial elongation and regression of the transient tripartite spines. However, the unique regeneration phase at 65 dph requires a more in-depth discussion on its potential biological or ecological significance.
- Comparing high and low-weight fish based on pond averages (group-level selection) introduces potential environmental bias. Acknowledge this limitation in the Discussion, noting that observed differences may not be purely due to individual genetic factors.
- The Conclusion appropriately calls for genetic improvement but should propose a specific follow-up study.
- Please propose validating the utility of the observed head coloration difference as a rapid, non-invasive early-stage selection tool for the better understanding for the readers.
Author Response
R1’s comments:
Comments 1: The Abstract misses the key finding such as the correlation between fast growth in red patches and slow growth in yellow patches on the head. Please incorporate this distinctive phenotypic marker into the Abstract as it is a major practical finding.
Response 1: Thank you for pointing this out. We agree with this comment. Therefore, we have added the relevant description in lines 27-29, lines 342-343, and lines 477-480.
Comments 2: The fertilization rate was derived from a small sample size of only 300 eggs. Please justify the representativeness of this subsample, if a larger population was used for the final rate.
Response 2: Thanks, we have corrected them as suggested in lines 96-102.
Comments 3: The authors found that C. sonnerati is slow-growing, the Introduction needs to strengthen the rationale for its culture.
Response 3: Thank you for pointing this out. Therefore, we have added the relevant description in lines 50-57.
Comments 4: There are inconsistencies in naming, such as "Somite formation stage" vs. "Muscle burl stage," between Table 1 and Figure 2. Ensure a consistent set of standardized scientific terminology is used across all figures, tables, text, etc. for the better transparency.
Response 4: Thank you for pointing this out. We have corrected them as suggested.
Comments 5: The discussion correctly notes the initial elongation and regression of the transient tripartite spines. However, the unique regeneration phase at 65 dph requires a more in-depth discussion on its potential biological or ecological significance.
Response 5: Thank you for pointing this out. Therefore, we have added the relevant description in lines 446-456.
Comments 6: Comparing high and low-weight fish based on pond averages (group-level selection) introduces potential environmental bias. Acknowledge this limitation in the Discussion, noting that observed differences may not be purely due to individual genetic factors.
Response 6: Thank you for pointing this out. We agree with this comment. Therefore, we have added the relevant content in lines 503-522.
Comments 7: The Conclusion appropriately calls for genetic improvement but should propose a specific follow-up study.
Response 7: Thank you for pointing this out. We have corrected them as suggested in lines 546-548.
Comments 8: Please propose validating the utility of the observed head coloration difference as a rapid, non-invasive early-stage selection tool for the better understanding for the readers.
Response 8: Thank you for pointing this out. We agree with this comment. Therefore, we have added the relevant description in lines 477-480.
Reviewer 2 Report
Comments and Suggestions for Authors
Materials and Methods
2.1.1 Broodstock composition
The manuscript does not report how many broodstock were used, nor the number of males and females. Please specify the broodstock population (sex ratio, total individuals, sizes) used for induced spawning.
L74 – Salinity units
The text reports salinity as “28–30%”. This is biologically unrealistic for seawater. Please clarify whether salinity is 28–30 ppt or 28–30 psu.
L71 – Number of incubation tanks and total eggs
The number of incubation tanks used and the total number of fertilized eggs at the beginning of the experiment are not reported. Please include the number of incubation tanks. Total initial number of eggs per tank or overall
L100. Origin of the 17,868 juveniles. Please clarify whether the 17,868 fish used for the 15-month grow-out originated from a single broodstock pair or multiple parents. This has implications for interpreting growth variation.
L120 – Statistical analysis (ANOVA assumptions & post hoc tests). Did the data meet the assumptions of normality and homogeneity of variance? It is unusual and unnecessary to apply two post hoc tests (LSD and Duncan) simultaneously. Please justify this choice or revise the statistical approach.
Results
Table 1 “Placenta” terminology. The term “placenta” is incorrect for teleost eggs. Please replace all occurrences with “blastodisc” or “germinal disc formation” to avoid confusion.
Figure 5 Given the continuous nature of growth, a fitted growth model (linear, or exponential etc) may better represent the data than multiple ANOVAs. Consider adding a growth curve with estimated parameters.
Section 3.4. Please specify the number of fish measured per pond for length and weight.
Discussion
Genetic vs environmental interpretation of growth variation. The manuscript concludes that growth variation “is linked to genetic factors,” yet there is no family structure, pedigree, or parentage information also No heritability estimates or genetic analyses were conducted. In addition, Environmental factors (early cannibalism, dominance hierarchy, feed adaptation rate, micro-environmental differences) are not accounted for. Please soften interpretations related to genetics and integrate discussion of environmental contributors to size heterogeneity.
L383 & L397. Tripartite spines and larval morphology This section is one of the strongest aspects of the manuscript. Strengthen it further by adding a short paragraph comparing C. sonnerati larval morphology to developmental patterns reported in other serranids/groupers.
Survival rates. The manuscript lacks survival data, which are critical in larval and juvenile rearing studies. Please report survival percentages (even approximate) at main stages: e.g. Hatch, 30 dph, 15-month grow-out, etc.
Cannibalism. Cannibalism is mentioned qualitatively, but no numerical estimates are given. If available, please include cannibalism rates or percentage losses attributed to cannibalism.
Growth parameters in tabular form. A table summarizing key growth parameters (e.g., TL, BW, SGR, condition factor, variance across ponds) would greatly improve clarity and readability of the growth results.
Comments on the Quality of English LanguageEnglish can be improved.
Author Response
R2’s comments:
Comments 1: The manuscript does not report how many broodstock were used, nor the number of males and females. Please specify the broodstock population (sex ratio, total individuals, sizes) used for induced spawning.
Response 1: Thank you for pointing this out. We agree with this comment. Therefore, we have added the relevant description in lines 87-93.
Comments 2: L74 – Salinity units. The text reports salinity as “28–30%”. This is biologically unrealistic for seawater. Please clarify whether salinity is 28–30 ppt or 28–30 psu.
Response 2: Thank you for pointing this out. We have corrected them as suggested in lines 100-102.
Comments 3: L71 – Number of incubation tanks and total eggs. The number of incubation tanks used and the total number of fertilized eggs at the beginning of the experiment are not reported. Please include the number of incubation tanks. Total initial number of eggs per tank or overall.
Response 3: Thank you for pointing this out. We have added the relevant description in lines 96-102.
Comments 4: L100. Origin of the 17,868 juveniles. Please clarify whether the 17,868 fish used for the 15-month grow-out originated from a single broodstock pair or multiple parents. This has implications for interpreting growth variation.
Response 4: Thank you for pointing this out. We agree with this comment. Therefore, we have added the relevant description in lines 125-126, and lines 129-130.
Comments 5: L120 – Statistical analysis (ANOVA assumptions & post hoc tests). Did the data meet the assumptions of normality and homogeneity of variance? It is unusual and unnecessary to apply two post hoc tests (LSD and Duncan) simultaneously. Please justify this choice or revise the statistical approach.
Response 5: Thank you for pointing this out. We agree with this comment. Therefore, we have revised the statistical approach in lines 156-163.
Comments 6: Table 1 “Placenta” terminology. The term “placenta” is incorrect for teleost eggs. Please replace all occurrences with “blastodisc” or “germinal disc formation” to avoid confusion.
Response 6: Thank you for pointing this out. We have replaced all occurrences of “placenta” with “blastodisc”.
Comments 7: Figure 5 Given the continuous nature of growth, a fitted growth model (linear, or exponential etc) may better represent the data than multiple ANOVAs. Consider adding a growth curve with estimated parameters.
Response 7: Thank you for pointing this out. We agree with this comment. Therefore, we have added the relevant content in lines 368-371, and Figure 5.
Comments 8: Section 3.4. Please specify the number of fish measured per pond for length and weight.
Response 8: Thank you for pointing this out. We have corrected them in lines 377-378.
Comments 9: Discussion. Genetic vs environmental interpretation of growth variation. The manuscript concludes that growth variation “is linked to genetic factors,” yet there is no family structure, pedigree, or parentage information also No heritability estimates or genetic analyses were conducted. In addition, Environmental factors (early cannibalism, dominance hierarchy, feed adaptation rate, micro-environmental differences) are not accounted for. Please soften interpretations related to genetics and integrate discussion of environmental contributors to size heterogeneity.
Response 9: Thank you for pointing this out. We agree with this comment. Therefore, we have added the relevant description in lines 503-522.
Comments 10: L383 & L397. Tripartite spines and larval morphology. This section is one of the strongest aspects of the manuscript. Strengthen it further by adding a short paragraph comparing C. sonnerati larval morphology to developmental patterns reported in other serranids/groupers.
Response 10: Thank you for pointing this out. We have added the relevant description in lines 446-456.
Comments 11: Survival rates. The manuscript lacks survival data, which are critical in larval and juvenile rearing studies. Please report survival percentages (even approximate) at main stages: e.g. Hatch, 30 dph, 15-month grow-out, etc.
Response 11: Thanks. We agree with this comment. Therefore, we have added the relevant content in lines 387-388, and lines 513-515.
Comments 12: Cannibalism. Cannibalism is mentioned qualitatively, but no numerical estimates are given. If available, please include cannibalism rates or percentage losses attributed to cannibalism.
Response 12: Thank you for pointing this out. We agree with this comment. Therefore, we have added the relevant description in lines 120-121, lines 123-125, and lines 510-513.
Comments 13: Growth parameters in tabular form. A table summarizing key growth parameters (e.g., TL, BW, SGR, condition factor, variance across ponds) would greatly improve clarity and readability of the growth results.
Response 13: Thank you for pointing this out. We agree with this comment. Therefore, we have added the relevant description in lines 150-155, lines 383-387, and Table 2.
Reviewer 3 Report
Comments and Suggestions for Authors
This study examines the embryonic and early post-embryonic development and growth characteristics of the tomato hind grouper (Cephalopholis sonnerati), a species that is new to aquaculture. Particular attention is paid to describing embryonic development and the morphological characteristics of larvae. A large number of illustrations accompany the manuscript, reflecting its content.
Describing the conditions required for farming fish species new to aquaculture is an important step in developing successful cultivation techniques. The data presented in this manuscript will contribute to the successful introduction of the tomato hind grouper to aquaculture, given that it demonstrates high rates of development and growth, and readily consumes artificial feed. A major positive aspect of the work is that the authors considered the full cycle of cultivation of this species, indicating the characteristics of its development and growth.
Overall, the work can be viewed positively. However, some minor inaccuracies should be pointed out. Correcting these will increase the significance of the research.
Specific comments
- The introduction should briefly explain why studying the embryonic development and growth of the planned aquaculture species is important. The authors emphasise the importance of studying embryonic development (lines 344–346) and the significance of morphological characteristics in larvae (lines 400–402). Reflecting these aspects in the introduction would improve the reader's understanding of the text, particularly if they are unfamiliar with the topic.
- To enhance the methodological rigour of the study, I suggest the authors specify the age, weight and quantity of the breeding stock used to produce the eggs in the 'Materials and Methods' section (lines 65–67).
- Table 2 (line 379) should include a reference for each fish species. While the authors do cite them in the text, including the references directly in the table would make it more informative.
- As the authors are focusing on comparing the degree-hour index for different species of grouper, I believe it would be appropriate to include this data for the tomato grouper in both the abstract and the conclusion.
Author Response
R3’s comments:
This study examines the embryonic and early post-embryonic development and growth characteristics of the tomato hind grouper (Cephalopholis sonnerati), a species that is new to aquaculture. Particular attention is paid to describing embryonic development and the morphological characteristics of larvae. A large number of illustrations accompany the manuscript, reflecting its content.
Describing the conditions required for farming fish species new to aquaculture is an important step in developing successful cultivation techniques. The data presented in this manuscript will contribute to the successful introduction of the tomato hind grouper to aquaculture, given that it demonstrates high rates of development and growth, and readily consumes artificial feed. A major positive aspect of the work is that the authors considered the full cycle of cultivation of this species, indicating the characteristics of its development and growth.
Overall, the work can be viewed positively. However, some minor inaccuracies should be pointed out. Correcting these will increase the significance of the research.
Response: Thank you very much for the positive comments.
Comments 1: The introduction should briefly explain why studying the embryonic development and growth of the planned aquaculture species is important. The authors emphasise the importance of studying embryonic development (lines 344–346) and the significance of morphological characteristics in larvae (lines 400–402). Reflecting these aspects in the introduction would improve the reader's understanding of the text, particularly if they are unfamiliar with the topic.
Response 1: Thank you for pointing this out. We agree with this comment. Therefore, we have added the relevant description in lines 67-77.
Comments 2: To enhance the methodological rigour of the study, I suggest the authors specify the age, weight and quantity of the breeding stock used to produce the eggs in the 'Materials and Methods' section (lines 65–67).
Response 2: Thank you for pointing this out. We have corrected them as suggested in lines 87-93.
Comments 3: Table 2 (line 379) should include a reference for each fish species. While the authors do cite them in the text, including the references directly in the table would make it more informative.
Response 3: Thanks. We agree with this comment. Therefore, we have added the relevant references in Table 2.
Comments 4: As the authors are focusing on comparing the degree-hour index for different species of grouper, I believe it would be appropriate to include this data for the tomato grouper in both the abstract and the conclusion.
Response 4: Thank you for pointing this out. We agree with this comment. Therefore, we have added the relevant description in lines 25-27, and lines 537-539.
Round 2
Reviewer 1 Report
Comments and Suggestions for Authors
The authors have done a good job, but the introduction section still needs good expansion with the latest citations. It is mandatory to extend the introduction section for the betterment of the manuscript.
Author Response
Comments 1: The authors have done a good job, but the introduction section still needs good expansion with the latest citations. It is mandatory to extend the introduction section for the betterment of the manuscript.
Response 1: Thank you for pointing this out. We agree with this comment. Therefore, we have added the latest citations in lines 62-65, and lines 74-84.
Reviewer 2 Report
Comments and Suggestions for Authors
I'm happy with the changes in the manuscript.
Author Response
Comments: I'm happy with the changes in the manuscript.
Response: Thank you very much for the positive comments.